# The diversity and function of sourdough starter microbiomes

Elizabeth A Landis[1†], Angela M Oliverio[2,3†], Erin A McKenney[4,5],
Lauren M Nichols[4], Nicole Kfoury[6], Megan Biango-Daniels[1], Leonora K Shell[4],
Anne A Madden[4], Lori Shapiro[4], Shravya Sakunala[1], Kinsey Drake[1],
Albert Robbat[6], Matthew Booker[7], Robert R Dunn[4,8], Noah Fierer[2,3],
Benjamin E Wolfe[1]*

[1]Department of Biology, Tufts University, Medford, United States; [2]Department of Ecology and Evolutionary Biology, University of Colorado, Boulder, United States; [3]Cooperative Institute for Research in Environmental Sciences, University of Colorado, Boulder, United States; [4]Department of Applied Ecology, North Carolina State University, Raleigh, United States; [5]North Carolina Museum of Natural Sciences, Raleigh, United States; [6]Department of Chemistry, Tufts University, Medford, United States; [7]Department of History, North Carolina State University, Raleigh, United States; [8]Danish Natural History Museum, University of Copenhagen, Copenhagen, Denmark

*For correspondence:
benjamin.wolfe@tufts.edu

[†]These authors contributed equally to this work

**Abstract** Humans have relied on sourdough starter microbial communities to make leavened bread for thousands of years, but only a small fraction of global sourdough biodiversity has been characterized. Working with a community-scientist network of bread bakers, we determined the microbial diversity of 500 sourdough starters from four continents. In sharp contrast with widespread assumptions, we found little evidence for biogeographic patterns in starter communities. Strong co-occurrence patterns observed in situ and recreated in vitro demonstrate that microbial interactions shape sourdough community structure. Variation in dough rise rates and aromas were largely explained by acetic acid bacteria, a mostly overlooked group of sourdough microbes. Our study reveals the extent of microbial diversity in an ancient fermented food across diverse cultural and geographic backgrounds.

## Introduction

Sourdough bread is a globally distributed fermented food that is made using a microbial community of yeasts and bacteria. The sourdough microbiome is maintained in a starter that is used to inoculate dough for bread production (*Figure 1A*). Yeasts, lactic acid bacteria (LAB), and acetic acid bacteria (AAB) in the starter produce $CO_2$ that leavens the bread. Microbial activities including the production of organic acids and extracellular enzymes also impact bread flavor, texture, shelf-stability, and nutrition (*Arendt et al., 2007*; *De Vuyst et al., 2016*; *Gobbetti et al., 2014*; *Hansen and Schieberle, 2005*; *Salim-ur-Rehman et al., 2006*). Starters can be generated de novo by fermenting flour and water or acquired as established starters from community members or commercial sources. Home-scale fermentation of sourdough is an ancient and historically important practice (*Cappelle et al., 2013*) that experienced a cultural resurgence during the COVID-19 pandemic (*Easterbrook-Smith, 2020*).

Despite being an economically and culturally significant microbiome, a comprehensive survey of sourdough starter microbial communities has not yet been conducted. Previous studies have primarily focused on starters from regions within Europe (*De Vuyst et al., 2014*; *Gänzle and Ripari, 2016*;

**eLife digest** Sourdough bread is an ancient fermented food that has sustained humans around the world for thousands of years. It is made from a sourdough 'starter culture' which is maintained, portioned, and shared among bread bakers around the world. The starter culture contains a community of microbes made up of yeasts and bacteria, which ferment the carbohydrates in flour and produce the carbon dioxide gas that makes the bread dough rise before baking.

The different acids and enzymes produced by the microbial culture affect the bread's flavor, texture and shelf life. However, for such a dependable staple, sourdough bread cultures and the mixture of microbes they contain have scarcely been characterized. Previous studies have looked at the composition of starter cultures from regions within Europe. But there has never been a comprehensive study of how the microbial diversity of sourdough starters varies across and between continents.

To investigate this, Landis, Oliverio et al. used genetic sequencing to characterize the microbial communities of sourdough starters from the homes of 500 bread bakers in North America, Europe and Australasia. Bread makers often think their bread's unique qualities are due to the local environment of where the sourdough starter was made. However, Landis, Oliverio et al. found that geographical location did not correlate with the diversity of the starter cultures studied. The data revealed that a group of microbes called acetic acid bacteria, which had been overlooked in past research, were relatively common in starter cultures. Moreover, starters with a greater abundance of this group of bacteria produced bread with a strong vinegar aroma and caused dough to rise at a slower rate.

This research demonstrates which species of bacteria and yeast are most commonly found in sourdough starters, and suggests geographical location has little influence on the microbial diversity of these cultures. Instead, the diversity of microbes likely depends more on how the starter culture was made and how it is maintained over time.

*Hammes et al., 2005*; *Minervini et al., 2014*) and the diversity of sourdough starters in North America is poorly characterized (*Kline and Sugihara, 1971*; *Sugihara et al., 1971*). Most previous studies have applied a range of culture-based techniques to characterize sourdough microbial diversity (*De Vuyst et al., 2014*; *Gänzle and Ripari, 2016*) making it difficult to understand distributions of sourdough bacterial and fungal taxa due to the variability and biases in these approaches. Sourdough starters are maintained in many households, but these starters have generally been overlooked in previous studies which have focused on large bakeries and industrial settings. Household starters are likely distinct from those found in bakeries due to a greater heterogeneity in environments, production practices, and ingredients.

Two major factors, geographic location and maintenance practices, are often invoked as major drivers of sourdough biodiversity. Sourdough communities in the same region may be similar in composition due to restricted dispersal of microbes or in response to regional microclimates. Producers often tout their breads' distinct regional properties, crediting the environment for unique bread characteristics. There is some evidence of local geographic structure of sourdough microbes (*Liu et al., 2018*; *Scheirlinck et al., 2007b*), but biogeographic patterns of sourdough diversity have not been quantified at a continental-scale. Likewise, experimental evidence from individual sourdoughs suggests that abiotic conditions (process parameters including differences in starter maintenance techniques, ingredients, and environmental conditions) and microbial interactions can impact starter community structure (*De Vuyst et al., 2014*; *Gänzle and Ripari, 2016*; *Minervini et al., 2014*; *Ripari et al., 2016*; *Van Kerrebroeck et al., 2017*), but the relative importance of these processes across diverse starters is unknown.

Through the collaborative power of a global network of community scientists, we collected 500 sourdough starters from across the world with dense sampling of the United States (United States = 429; Canada = 29; Europe n = 24; Australia and New Zealand = 17; and Thailand = 1; *Figure 1B*) to comprehensively characterize the microbial diversity of household sourdough starters. These starters varied greatly in their reported ages and maintenance histories (*Figure 1C–G*). Through both cultivation-dependent and cultivation-independent methods, we revealed the

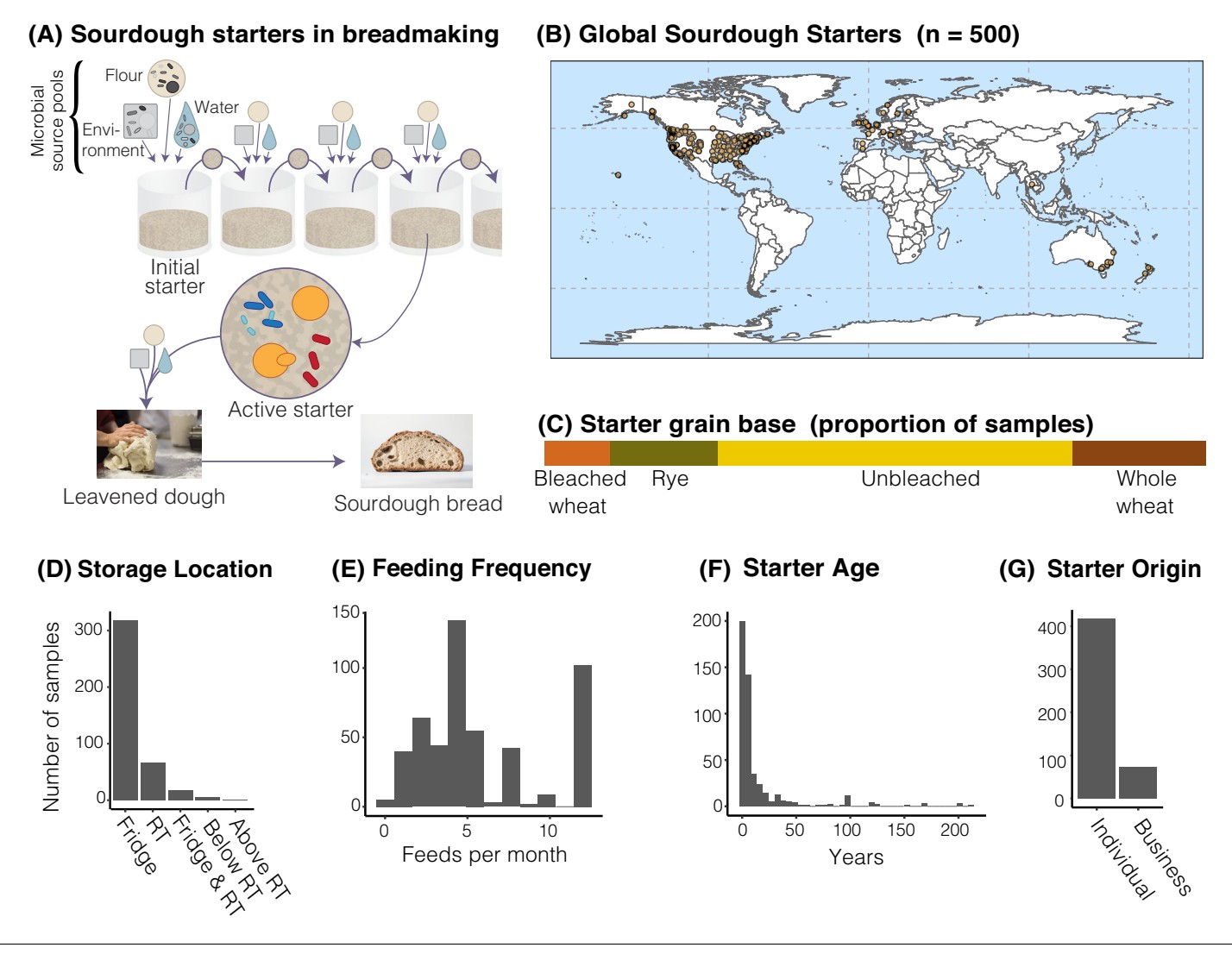

**Figure 1.** The distribution of sourdough starters sampled in this study. (A) Overview of the process of serial transfer of a sourdough starter. (B) Locations of the 500 sourdough starters analyzed in this study. Each dot represents one sourdough starter. (C-G) Characteristics of collected sourdough starters. In (D), RT = room temperature. In (G), 'Individual' = participant reported acquiring their starter from another individual (not a commercial source); 'Business' = participant reported acquiring their starter from a commercial source.

ecological distributions of widespread sourdough yeasts and bacteria. By analyzing a broad suite of starter metadata, our intensive sampling identified the roles of geography and process parameters in shaping starter diversity.

Using synthetic sourdough communities, we identified a dynamic network of species interactions within sourdough microbiomes that helps explain the distributions of major yeasts and bacteria. We also determined linkages between sourdough starter microbial diversity and baking-relevant functions including the rate of dough rise and volatile organic compound (VOC) production. This study is the first to combine a large-scale survey of sourdough starter microbial diversity with quantitative analysis of the factors that shape the composition and function of starter microbiomes.

## Results

### Diversity of sourdough starters

We first identified the microbial communities of sourdough starters by 16S and ITS rRNA gene amplicon sequencing of samples that were shipped to us and frozen upon arrival. When considering both fermentation-relevant microbes (yeasts, LAB, and AAB) as well as other microbial taxa, each starter sample contained a median of seven bacterial and 35 fungal amplicon sequence variants (ASVs). LAB (order: Lactobacillales) and AAB (order: Rhodospirillales) together comprised over 97% of bacterial reads (per sample mean), with yeasts (order: Saccharomycetales) comprising over 70% of fungal reads (*Figure 2—figure supplement 1*, *Figure 2—source data 1*, *2*). The other fungi and bacteria detected were common indoor and outdoor molds, plant pathogens, and plant endophytes as well as microbes associated with human skin, drinking water, and soil. Unless otherwise indicated, we did not include these environmental microbes in our further analyses because of their limited roles in sourdough fermentation.

Sourdough communities exhibited consistent patterns of strong species dominance or co-occurrence (*Figure 2A*). Many communities were dominated by a single yeast and/or bacterial species with a median of three LAB/AAB and one yeast per starter (*Figure 2A*-*Figure 2—figure supplement 2*). For example, *Saccharomyces cerevisiae* accounted for >50% of fungal ITS reads in 77% of samples. The LAB *L. sanfranciscensis* was the dominant bacterium in most sourdoughs where it occurred and was negatively associated with the widespread *L. plantarum* and *L. brevis* (p<0.001; *Figure 2A*, *Figure 2—figure supplement 1*, *Figure 2—source data 3*). The LAB *Lactobacillus plantarum* and *L. brevis* were the most commonly observed pair of co-occurring taxa (in 177 of 500 starters, p<0.001; *Figure 2A*-*Figure 2—figure supplement 3*). Interactions predicted in the literature, including *L. sanfranciscensis:Kazachstania humilis* co-occurrence (*Brandt et al., 2004*; *De Vuyst et al., 2016*) and *L. sanfranciscensis:S. cerevisiae* co-exclusion (*Gobbetti et al., 1994*), were supported by in situ patterns of diversity (*L. sanfranciscensis:K. humilis* p<0.01, *L. sanfranciscensis:S. cerevisiae* p=0.01).

One striking pattern across our dataset was the highly variable abundance of AAB across individual starters. These bacteria have been reported in sourdough (*Minervini et al., 2014*; *Ripari et al., 2016*), but are generally understudied as indicated by their almost complete absence in many key reviews of sourdough microbial diversity (*De Vuyst et al., 2014*; *Gänzle and Ripari, 2016*; *Van Kerrebroeck et al., 2017*). In our sample set, 147 starters contained AAB (>1% relative abundance) including *Acetobacter*, *Gluconobacter*, or *Komagataeibacter* species (*Figure 2A*, *Figure 2—source data 2*). AAB require specialized culture conditions (*Kim et al., 2019*) and cultivation biases in previous studies (*De Vuyst et al., 2014*; *Gänzle and Ripari, 2016*; *Van Kerrebroeck et al., 2017*) may explain their widespread omission from our understanding of sourdough biodiversity.

### Geography, process parameters, and abiotic factors are poor predictors of sourdough starter microbiome composition

We first examined whether sourdough starter community composition was correlated with geographic distance between starters using a distance-decay analysis. Across the continental U.S. where we had the highest sample density, taxonomic composition was not correlated with geographic distance (Mantel ρ = 0.0, p>0.05 for both LAB/AAB and yeasts). At the global-scale, yeast taxonomic composition was weakly predicted by geography (Mantel ρ = 0.07, p<0.01). The geographic signal was stronger when all fungal ASVs were included (Mantel ρ = 0.23, p≤0.001 globally), potentially due to differences in the non-yeast fungi (molds, plant pathogens) found within local ingredients and/or environments (*Barberán et al., 2015*).

While differences in the overall composition of sourdough starter communities were not correlated with geographic distance between starters, species of fermentation-relevant bacteria or yeast may be enriched in some geographic regions due to dispersal or production processes. To determine if there are sourdough microbial species which are restricted to particular regions of the U.S., we used k-means clustering to group samples at two scales: k = 4 (larger regions, *Figure 2B*) and k = 15 (smaller geographic regions, *Figure 2C*). Next, we identified indicator taxa that were significantly enriched in these regions. Indicator species analysis detects individual taxa that are enriched under particular conditions, where indicator strength ranges from 0 to 1. An indicator strength above 0.25 is traditionally classified as a species that is strongly associated with a condition

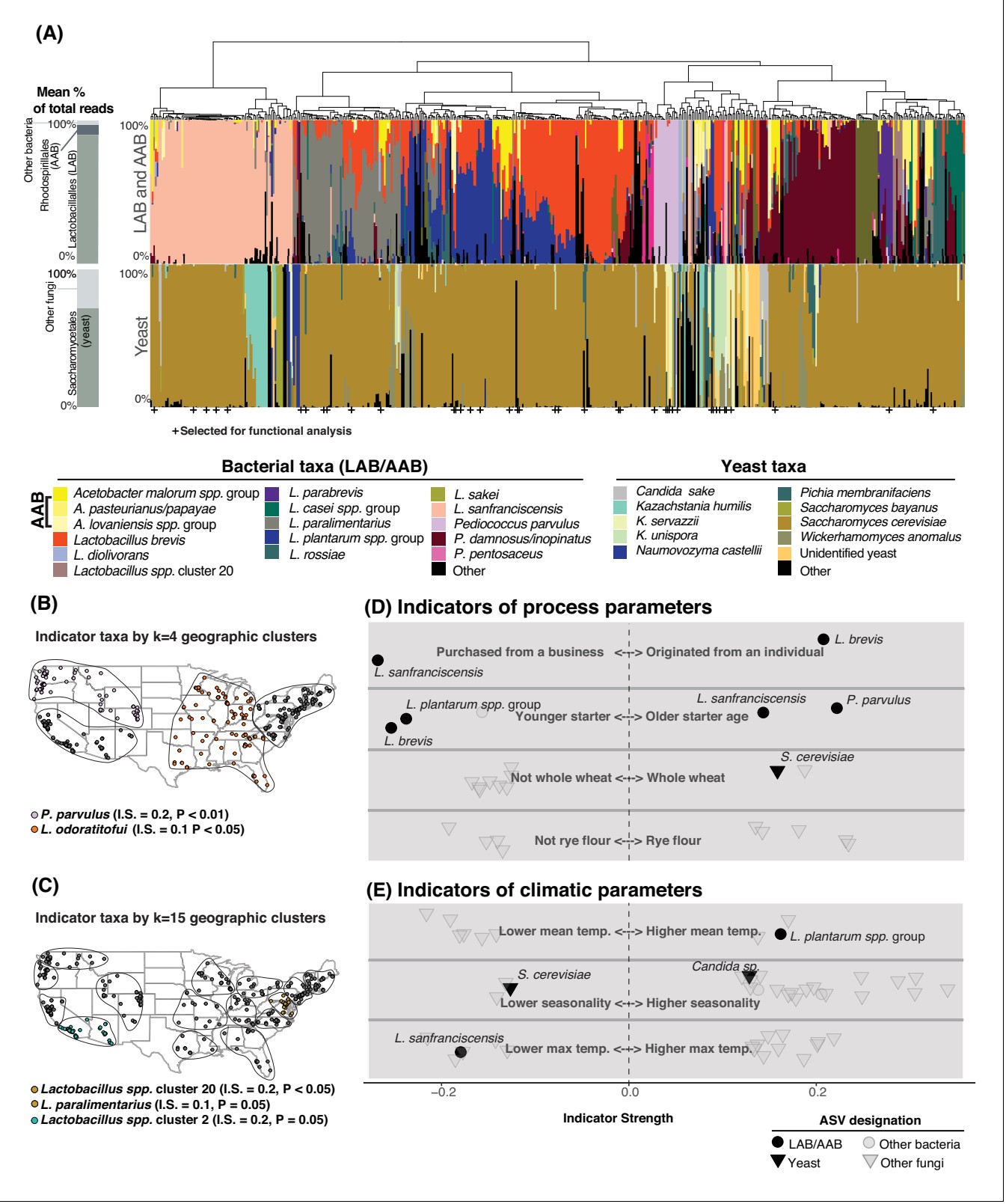

**Figure 2.** Process parameters and geography weakly predict the diversity of sourdough starters. (**A**) Starters (n = 500) hierarchically clustered by Bray-Curtis dissimilarities. The stacked bar chart on the left shows the proportion of total reads across all samples belonging to the orders Rhodospirillales (AAB), Lactobacillales (LAB), and Saccharomycetales (yeast) (see *Figure 2—source data 1*, *2* for a complete list of these taxa). On the right, each column represents an individual sourdough starter. See *Figure 2—source data 3* for co-occurrence analysis results. Below the barchart, + indicates

*Figure 2 continued on next page*

*Figure 2 continued*

samples selected for functional analysis (*Figure 4*). Continental U.S. geographic regions were clustered at two scales: k = 4 (**B**) and k = 15 (**C**). Dots represent individual samples. Each geographic cluster is encircled. Colored dots represent clusters where indicator taxa were significantly (p<0.05) associated with geographic clusters according to indicator species analysis. In (**D**) and (**E**), indicator strengths (*Figure 2—source data 6*) illustrate individual ASVs that are significantly associated with (**D**) process parameters including starter maintenance techniques and (**E**) climatic parameters. Each individual dot or triangle represents an individual ASV of bacterium or fungus, respectively.

The online version of this article includes the following source data and figure supplement(s) for figure 2:

**Source data 1.** The most abundant bacterial and fungal taxa across the 500 sourdough starter samples that are not typically considered an active part of starter communities (e.g. excluding yeasts, lactic acid bacteria, and acetic acid bacteria).

**Source data 2.** The most abundant yeast, lactic acid bacteria, and acetic acid bacteria species across the 500 sourdough starter samples.

**Source data 3.** Co-occurrence statistics of sourdough yeasts and bacteria calculated with the R package 'cooccur'.

**Source data 4.** Predictors (n=33) included in PERMANOVA tests on bacterial and fungal dissimilarities.

**Source data 5.** Abiotic properties are poor predictors of overall variation in both bacterial and fungal community composition across sourdough starters.

**Source data 6.** Complete list of indicator taxa and summary statistics, as described in *Figure 2*.

**Figure supplement 1.** Phylogenetic trees of (**A**) lactic acid bacteria (LAB) and (**B**) acetic acid bacteria (AAB) detected in the 500 sourdough starters.

**Figure supplement 2.** Richness across starter microbial communities.

**Figure supplement 3.** A co-occurrence analysis showing all significant associations.

**Figure supplement 4.** Geographic location is a weak predictor of fungal sourdough starter community and not a significant predictor of bacteria.

---

(*Dufrêne and Legendre, 1997*; *Gebert et al., 2018*). We detected several taxa enriched within regions of the continental U.S., although indicator strength for each of these was weak ($\leq$0.2; *Figure 2B–C*). Collectively, the distance-decay and indicator species analyses demonstrate a limited role of geography in structuring the taxonomic diversity of sourdough microbial communities (*Figure 2B–C-Figure 2—figure supplement 4*).

We next tested whether 33 other types of metadata collected for each starter could predict the observed composition of starters; these factors included age of starter, storage location, feed frequency, grain input, home characteristics, and climatic factors (*Figure 2D–E*; *Figure 2—source data 4*). Together, these predictors accounted for less than 10% of the variation in community composition for both bacterial and fungal communities in both the U.S. and global datasets (PERMANOVA $R^2$ of all bacterial ASVs = 8.3% and $R^2$ of all fungal ASVs = 7.5% of the overall variation in Bray-Curtis dissimilarities for the global dataset; $R^2$ bacteria = 9.0% and fungi = 5.0% for the U.S.; *Figure 2D–E*, *Figure 2—source data 5*).

Some fermentation-relevant taxa were enriched under particular conditions (*Figure 2D–E*, *Figure 2—source data 6*). For example, younger starters were often dominated by the LAB *L. plantarum* (indicator strength (IS) = 0.238, p<0.001) and *L. brevis* (IS = 0.254, p<0.001), while older starters often contained *L. sanfranciscensis (IS = 0.043, p=0.01)* and *P. parvulus* (IS = 0.222, p<0.001; *Figure 2D*, *Figure 2—source data 6*). Previous studies have not found strong associations between flour type or other fermentation practices and yeast species present (*De Vuyst et al., 2014*; *Vrancken et al., 2010*). In our study, *S. cerevisiae* was weakly associated with starters whose grain base was whole wheat (IS = 0.157, p=0.04). Most of the other fungal indicator species were non-yeast molds and plant endophytes, which were enriched under particular climatic conditions (*Figure 2E*, *Figure 2—source data 6*). No AAB taxa were enriched under any particular fermentation practice or climatic condition.

The history and origins of sourdough starters may also explain the distribution of some widespread microbial species. Sourdough bakers can either begin their starters de novo from flour and water or obtain an established starter from a business or individual. The LAB *L. brevis* was associated with de novo starters (IS = 0.206, p=0.04). There were 73 starters in our collection that were originally acquired by home bakers from a bakery or other commercial source and *L. sanfranciscensis* was abundant in these commercial starters (IS = 0.267, p=0.04). This suggests that *L. sanfranciscensis* thrives under commercial production conditions and has been widely distributed among bakers, where it persists in home fermentations.

## Ecological distributions of sourdough microbes are structured by biotic interactions

We next determined whether individual growth rate and/or biotic interactions among taxa could help to explain distributions of sourdough species (*Friedman et al., 2017*; *Vega and Gore, 2018*). Whereas previous studies have focused on single pairs of interacting sourdough microbes (*De Vuyst and Neysens, 2005*; *Gobbetti et al., 1994*; *Sieuwerts et al., 2018*), we chose eight isolates: four LAB and four yeasts, representing the most frequent yeasts and bacteria in sourdough that also displayed strong positive and/or negative patterns of co-occurrence (*Figure 3A*, *Figure 2—source data 2*, *3*). We did not include AAB in these interaction experiments because they were not significantly associated with other microbial taxa in the amplicon dataset. To determine the growth ability of each species alone, we measured colony forming units at the end of six 48 hr transfers in a liquid, cereal-based fermentation medium (*Figure 3A*). To determine competitive ability, we serially passaged 1:1 mixtures of each pair of the eight microbial species through this medium for six 48 hr transfers and assessed the relative abundance of each isolate in each pair (*Figure 3A–B-Figure 3—figure supplement 1*). We determined whether each of the eight species could co-persist in pairwise competitions, where co-persistence was defined as both isolates being present at >1% relative abundance after the six transfers.

Taxa that are able to reach high cell densities in the sourdough environment may be able to outcompete many other sourdough species. When comparing the growth of each isolate alone over six transfers to its ability to persist in pairwise competitions, we found a significant positive relationship (Spearman's ρ = 0.81, p<0.05, *Figure 3C*). For example, the LAB *L. brevis* is able to reach high densities when grown alone in our synthetic sourdough environment and is also able to persist when paired with all seven competing LAB and yeast species. In contrast, the LAB *L. sanfranciscensis* has one of the lowest densities when grown alone and is only able to persist when grown with the yeast *K. humilis*. We did not detect a significant correlation between the ability to persist in pairwise competitions and frequency of each taxon across the amplicon sequencing dataset (Spearman's ρ = 0.39, p>0.05, *Figure 3D*), or between growth alone and frequency in the amplicon sequencing dataset (Spearman's ρ = 0.57, p>0.05).

To test how well specific interactions among yeast and LAB detected in sourdoughs could be recapitulated in our in vitro system, we compared significant co-occurrence patterns inferred from amplicon sequence data (positive or negative associations), with synthetic co-persistence patterns from the competition experiments. Of the 16 significant associations detected in the amplicon dataset, most (14 out of 16) were within-kingdom interactions (yeast:yeast and bacteria:bacteria) and only two were cross-kingdom interactions: a negative pattern of co-occurrence between *Saccharomyces cerevisiae* and *Lactobacillus sanfranciscensis*, and a positive pattern of co-occurrence between *Kazachstania servazzii* and *Pediococcus damnosus*. In the competition experiment, yeasts and bacteria co-persisted with each other in half of all pairings (8 of 16), and within-kingdom (yeast:yeast or bacteria:bacteria) species pairs co-persisted in 3 of 12 pairings (*Figure 3B and E*). Most species pairs (20 of 28) did not show significant positive or negative associations in the amplicon dataset. For the eight significant co-occurrence interactions detected in the amplicon sequence dataset, seven out of eight were recapitulated in our synthetic communities (*Figure 3E*; p<0.05). The consistency in directionality between pairwise co-occurrence patterns observed between the 500 sourdough starters and in vitro suggests robust microbial interactions in sourdough despite differences in environmental conditions and fermentation practices.

## Microbial composition influences dough rise and aroma profiles

To determine how variation in starters' community composition impacts their functional attributes, we selected a subset of 40 starters that spanned the spectrum of sourdough diversity (*Figure 2A*). We measured two baking-relevant functions: emissions of volatile organic compounds (VOCs), which can impact baked sourdough bread aromas (*Pétel et al., 2017*), and leavening (measured as rate of dough rise), which can impact bread structural properties (*Arendt et al., 2007*).

Across all samples, 123 volatile compounds were detected by GC/MS including well-known sourdough compounds 3-methyl-1-butanol, ethyl alcohol, acetic acid, and ethyl acetate (median number of compounds detected per starter = 85; *Figure 4—figure supplement 1*). Sensory analysis yielded 14 dominant notes across the 40 starters, including yeasty, vinegar/acetic acid/acetic sour, green

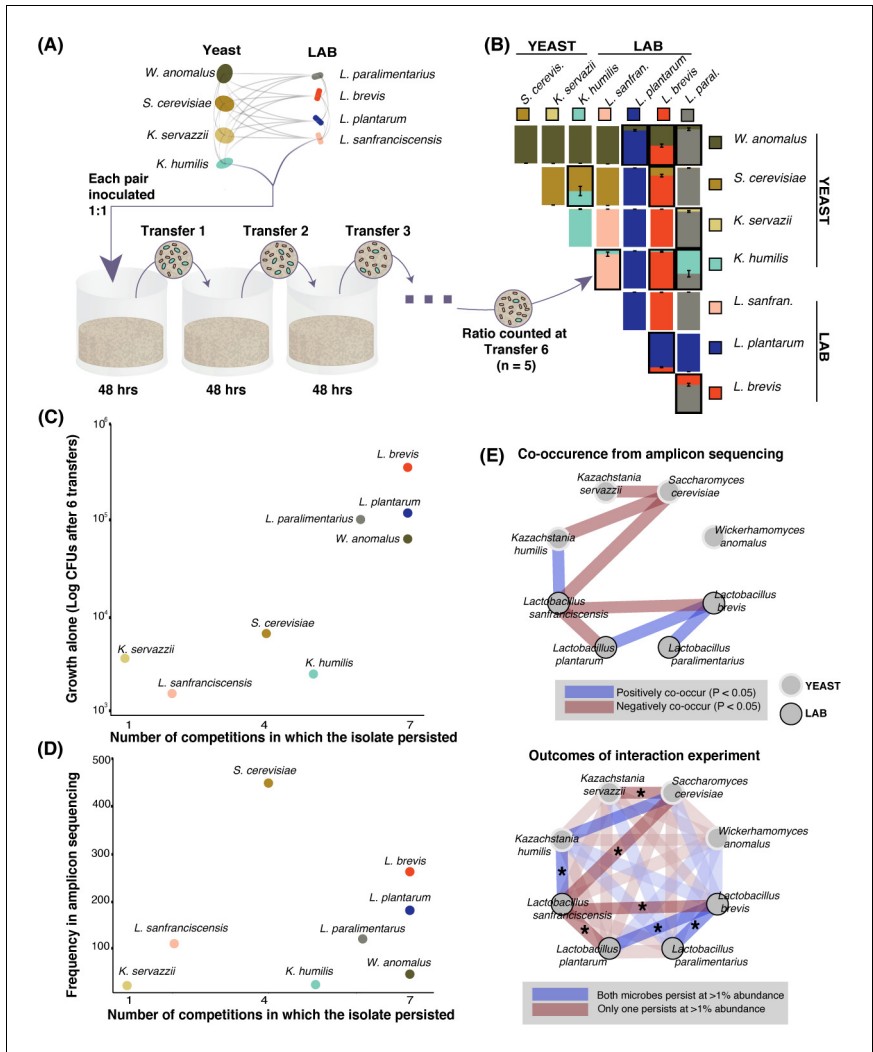

**Figure 3.** Growth rate and competitiveness fail to explain abundance patterns, but co-occurrence patterns in situ are recovered in pairwise coexistence experiments. (**A**) All possible species 1:1 pairs were grown in 200 µL liquid flour media (n = 5) and 10 µL was serially transferred every 48 hr. This conceptual schematic follows one pairing, *K. humilis* and *L. sanfranciscensis,* to illustrate the experimental approach. (**B**) Mean relative abundance of pairs at the end of transfer six. Pairs where both isolates persisted (>1% relative abundance) at the end of the experiment are outlined; error bars are ± SE. For all replicates at transfers one, three, and six, see *Figure 3—figure supplement 1.* (**C**) Correlation between growth of individual isolates alone (CFUs of each isolate after six transfers) and a simple persistence index (the number of competitions where the isolate persisted) found a positive and significant relationship (detection limit of mean one percent abundance across replicates; Spearman's ρ = 0.81, p=0.02). (**D**) Frequency of each taxon in the amplicon sequencing dataset and the number of competitions where that isolate persisted was positively associated, but not significant (Spearman's ρ = 0.39, p=0.34). (**E**) Significant (Bonferroni-corrected p<0.05) patterns of co-occurrence between taxa in our amplicon sequencing (top) were replicated 7 of 8 times in our experimental manipulation (bottom). All pairwise experimental outcomes from transfer six are represented in the bottom part of the figure; the eight pairs that have significant co-occurrence associations are highlighted and the experimental outcomes that matched the co-occurrence data have an asterisk. Refer to *Figure 2—figure supplement 3- Figure 2—source data 3* for all amplicon co-occurrence data.

The online version of this article includes the following source data and figure supplement(s) for figure 3:

**Source data 1.** CFU counts and relative abundance data from competitions, transfers one, three, and six.
**Figure supplement 1.** Pairwise competition experimental outcomes at transfers one, three, and six.

apple, fermented sour, and ethyl acetate/solvent (*Figure 4—figure supplement 3*, *Figure 4—source data 2*). The source of the starter inoculum explained most of the variation in VOC profile dissimilarities (PERMANOVA $R^2$ = 91% and p≤0.001), differences in maximum dough height (adjusted $R^2$ = 22%, ANOVA p<0.05), and dough rise rates (adjusted $R^2$ = 42%, ANOVA p<0.001; *Figure 4—figure supplements 1–2* and *Video 1*). This demonstrates wide variation in functional capacities across the 40 starters and low variation across experimental replicates within one starter. Starter community composition did not correlate with the dominant sensory note assigned by the sensory panel (PERMANOVA $R^2$ = 37%, p=0.16). While total bacterial and yeast species richness was not significantly correlated with VOC richness (Spearman's ρ = −0.09, p>0.05), starter microbial community composition was significantly correlated with VOC profiles (Bray-Curtis dissimilarity for community composition and VOC profiles; Mantel ρ = 0.19, p≤0.01).

When we determined whether VOC variation was driven by particular sourdough species, only the *Acetobacter malorum* spp. group emerged as significant (ρ = 0.528, FDR-corrected p<0.05; *Figure 4—source data 1*). More generally, total % AAB explained was strongly correlated with variation in VOCs (Mantel ρ = 0.73, p<0.001; *Figure 4*) and sensory notes (Kruskal-Wallis p<0.05; *Figure 4*), with the differences most strongly tied to the vinegar note compared to green apple and fermented sour notes (Dunn test p=0.04 and 0.06 respectively; *Figure 4—figure supplement 3*). The total relative abundance of AAB was also negatively correlated with the rate of dough rise (ρ = −0.51, p<0.001; *Figure 4*), which may be explained by the production of compounds such as acetic acid and inhibition of yeast or LAB growth in sourdough starters. The combined functional datasets highlight that variation in AAB abundances is a key driver of functional diversity across our sourdough collection.

## Discussion

This study presents the first intercontinental atlas of sourdough starter microbial communities. Our unprecedented scale of sampling demonstrates how sourdough maintenance and acquisition practices as well as microbial interactions can impact the biodiversity of starters. We also reveal novel structure-function linkages in this ancient fermented food. In combination with thousands of years of traditional knowledge about how to make good bread, these results provide possible management strategies for manipulating starter diversity.

Contradicting widespread beliefs about the regionality of sourdough microbiomes (e.g. the famous 'San Francisco sourdough'), our comprehensive sampling demonstrates that geographic location does not determine sourdough microbial composition. Previous studies using limited sampling have suggested that sourdough starters can vary across geographic regions (*Liu et al., 2018*; *Scheirlinck et al., 2007b*), but we are unaware of other studies that have rigorously explored sourdough microbiome biogeographic patterns in a distance-decay framework. The limited role of geography in explaining sourdough diversity may be driven by the widespread movement of starters across large geographic distances through starter sharing or commercial distribution. Flour, a major potential source of microbes in de novo starters (*Minervini et al., 2015*; *Reese et al., 2020*), is also moved across large spatial scales. This geographic homogenization of starter and flour microbes likely swamps out any regional

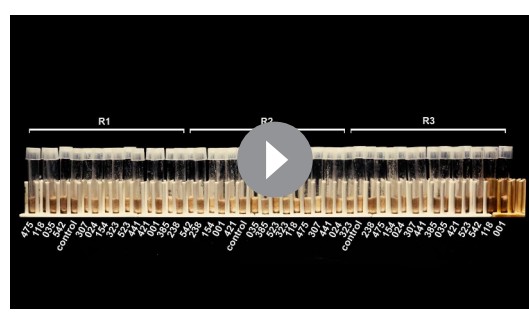

**Video 1.** Dough rise analysis using a common garden sourdough starter approach. Video shows the first of three batches of sterilized flour and water (n = 40, three replicates of each) that were inoculated with sourdough starters. Dough rise was measured by tracking the tops of each dough using video tracking software over the course of 36 hrs. Changes from the starting position were fitted with logistic growth curves using the R package GrowthCurver. Videos for each tube were trimmed if they fell to more than 5% of their maximum rise value. Dough rise rates ranged from 0.1 to 1.5 mm/hr. For scale, tubes are 103 mm tall with their caps. Doughs were removed part way through for placement of volatile organic compound collection bars which were present during hours 12–36. Video also available at: https://youtu.be/iK4lyRw2acA.
https://elifesciences.org/articles/61644#video1

differences in potential yeasts or bacteria that can disperse into starters.

Biogeographic patterns of sourdough microbes may exist at finer-scale resolutions than those used in the current study. We used amplicon sequencing to assess taxonomic diversity of sourdough because we were interested in species-level distributions of sourdough microbes. We acknowledge that intraspecific strain diversity plays important roles in many fermented foods (*El Khoury et al., 2017*; *Niccum et al., 2020*; *Walsh et al., 2017*), that microbial strains may have unique biogeographic patterns (*Gayevskiy and Goddard, 2012*), and that amplicon sequencing does not typically have the resolution to fully capture strain-level diversity (*Turaev and Rattei, 2016*). However, previous studies of other ecosystem types using similar amplicon sequencing approaches and at similar spatial scales have found strong distance-decay relationships of both fungal and bacterial communities (*Finkel et al., 2012*; *Ma et al., 2017*; *Talbot et al., 2014*). Future work using shotgun metagenomic sequencing or whole-genome sequencing of widespread sourdough microbes may reveal more nuanced biogeographic patterns in sourdough biodiversity.

Relative to bakeries, home producers likely follow less strict and sustained regimens for starter maintenance; they may vary techniques over time or keep imperfect records in regard to reported metadata. Still, process parameters explained some of the variation we detected through amplicon sequencing, with individual taxa being significantly associated with certain parameters. For example, *L. sanfranciscensis* was prevalent in older starters and starters purchased by participants from a business, which supports the hypothesis that it is selected for by bakers over generations of sourdough production (*Gänzle and Ripari, 2016*). Future studies that explicitly test how starter composition changes over time and across geographies in home fermentations are needed to better understand selection and stability in these ecosystems.

The concordance between our amplicon sequencing and competition experiment suggest that commonly or uncommonly observed species pairs may be due to complementary or competitive inter-species interactions. An important caveat is that the single representative isolates used in these experiments do not capture strain-level genomic and metabolic diversity, which has been shown to produce different competitive outcomes among and between strains of these sourdough taxa (*Rogalski et al., 2020*). Additionally, the pairwise interaction assay that we used does not capture the potential for interactions between three or more species. Though pairwise competitions have been shown to be predictive of higher-order interactions in simple microbial ecosystems (*Friedman et al., 2017*), outcomes of interactions in more complex synthetic sourdough communities with three or more species may differ from what we observed here. Future studies that use 'leave-one-out' approaches can identify potential roles of multidimensional microbial interactions in sourdough microbiomes.

We observed many striking microbial interactions in our synthetic system that were not predicted by sequencing. For example, *W. anomalus* was a strong competitor in vitro in this and other studies (*Daniel et al., 2011*; *Vrancken et al., 2010*), but it is not frequently found across the starters we sampled and co-occurred with other yeast species in the sourdoughs that were sequenced. This work adds to an existing body of evidence suggesting that *W. anomalus* is strongly competitive in sourdough, but may be dispersal limited, and/or that environmental conditions or multispecies interactions in home sourdough starters mitigate its competitive performance in situ.

Our integrative functional analysis of starter microbiomes highlights the disproportionate effects of AAB in shaping the sensory and leavening properties of sourdough. These bacteria have been historically underappreciated in most studies of sourdough microbes (*De Vuyst et al., 2014*; *Gänzle and Ripari, 2016*; *Van Kerrebroeck et al., 2017*). The limited use of media that can selectively grow AAB and the lack of metagenomic approaches in previous studies are two potential explanations for the underestimation of AAB abundances in the literature. Based on our findings, we argue that the relative abundance of this group should be considered a key factor in predicting the functional attributes of sourdough. However, the influence of AAB on sensory properties and dough rise is not straightforward and the correlation between percentage of AAB and vinegar sensory properties as well as slower dough rise rates was not seen in all samples. Further studies are needed to understand how VOCs produced by AAB contribute to the quality of baked sourdough bread.

Sourdough starters exemplify a unique type of microbial community: a top-down engineered microbiome that is stably maintained for many years. As our co-occurrence analysis and synthetic community validation revealed, these simple countertop microbial ecosystems provide ample opportunities to identify processes that structure microbiomes. Further experimental approaches that

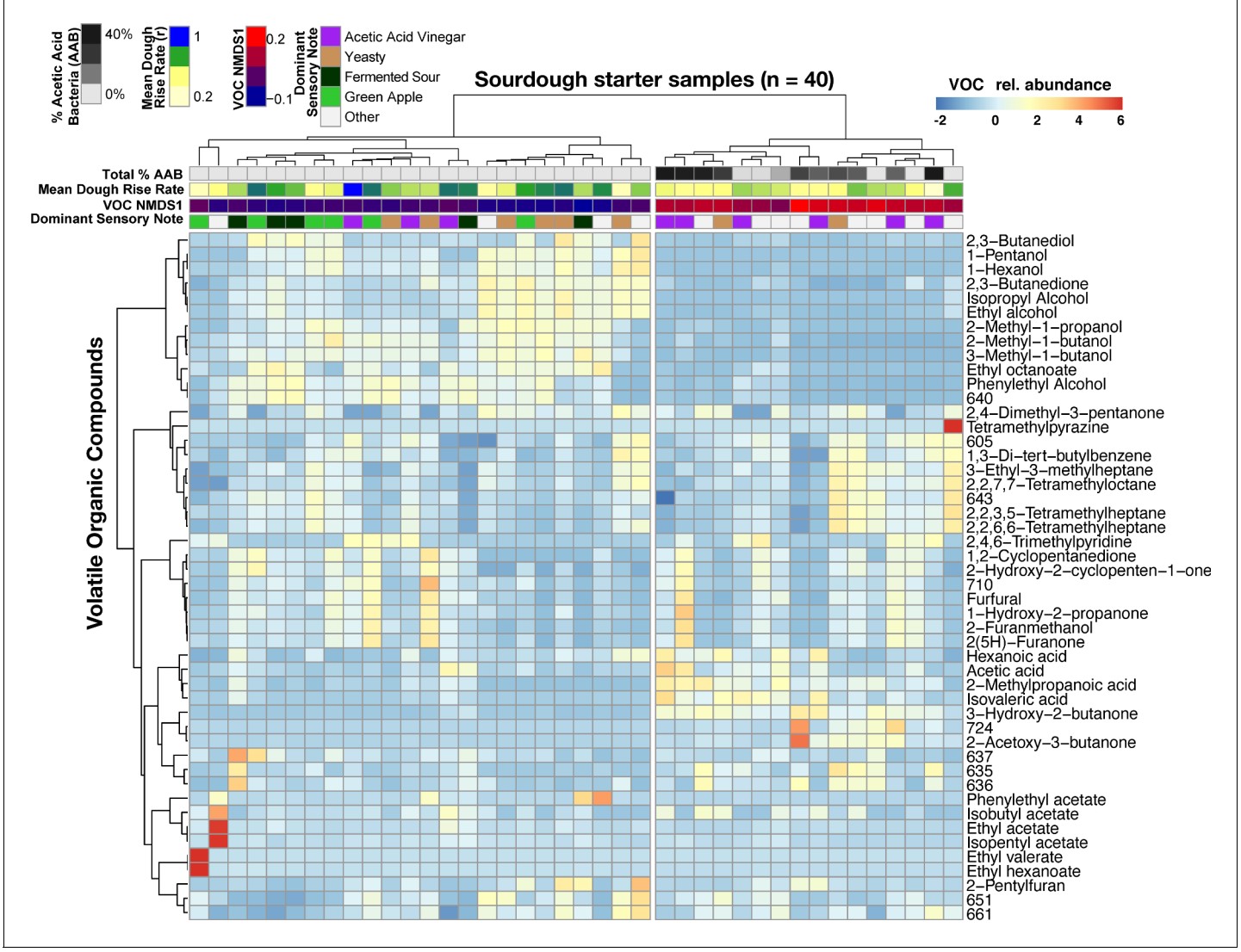

**Figure 4.** Acetic acid bacteria are drivers of sourdough starter functional diversity. Heatmap shows the relative abundances of VOCs (z-scores) across samples. Columns represent the 40 starter samples clustered with Bray-Curtis dissimilarities of VOC profiles, resulting in two main clusters. Rows show the top 48 VOCs clustered by correlation similarity. Numbered VOCs are unknown compounds. Top rows indicate the total percentage of AAB and the three measured functional outputs. Functional outputs were all predicted by % AAB including: (1) mean dough rise rate ($\rho = -0.51$, $p<0.001$), (2) the overall VOC composition represented by the first NMDS axis (see *Figure 4—figure supplement 1*; Mantel $\rho = 0.73$, $p<0.001$) and (3) the dominant sensory note (adj. $R^2 = 37\%$, $p<0.01$, see *Figure 4—source data 2* for all sensory notes).

The online version of this article includes the following source data and figure supplement(s) for figure 4:

**Source data 1.** The relationships between microbial taxa (lactic acid bacteria, acetic acid bacteria, and yeast) and functional outputs.

**Source data 2.** Complete list of sensory panel notes.

**Source data 3.** Dough rise data over the course of 36 hours of rise.

**Source data 4.** Volatile organic compound profiles collected for a subset of 40 starters.

**Figure supplement 1.** VOC data across replicate sourdough starters.

**Figure supplement 2.** Dough rise rates are predicted by starting microbial inoculum (adj. $R^2 = 0.42$, $p<0.001$).

**Figure supplement 3.** The four most frequently reported sensory notes from the 40 samples analyzed by an expert sensory panel.

manipulate other dimensions of sourdough, including phages, genomic heterogeneity, and evolutionary dynamics, will continue to uncover mechanisms of microbiome assembly in this ancient fermented food.

# Materials and methods

## Key resources table

| Reagent type (species) or resource | Designation | Source or reference | Identifiers | Additional information |
|---|---|---|---|---|
| Commercial assay or kit | Powersoil | Qiagen | Cat No./ID: 47014 | |
| Sequence-based reagent | 515 f | *Caporaso et al., 2011* | PCR primer | Forward primer used for amplifying bacterial DNA for amplicon sequencing |
| Sequence-based reagent | 806 r | *Caporaso et al., 2011* | PCR primer | Reverse primer used for amplifying bacterial DNA for amplicon sequencing |
| Sequence-based reagent | ITS1f | *Gardes and Bruns, 1993* | PCR primer | Forward primer used for amplifying fungal DNA for amplicon sequencing and Sanger sequencing |
| Sequence-based reagent | ITS2 | *White et al., 1990* | PCR primer | Reverse primer used for amplifying fungal DNA for amplicon sequencing |
| Sequence-based reagent | ITS4 | *White et al., 1990* | PCR primer | Reverse primer used for amplifying fungal DNA for Sanger sequencing |
| Sequence-based reagent | 27 f | *Lane, 1991* | PCR primer | Forward primer used for amplifying bacterial DNA for amplicon sequencing and Sanger sequencing |
| Sequence-based reagent | 1492 r | *Turner et al., 1999* | PCR primer | Reverse primer used for amplifying bacterial DNA for amplicon sequencing and Sanger sequencing |
| Software, algorithm | Dada2 | *Callahan et al., 2016* | | Software package for identifying amplicon sequence variants (ASVs) |
| Software, algorithm | raxml-HPC | *Stamatakis, 2014* | | Phylogenetic tree builder for taxonomic assignments of ASVs |
| Software, algorithm | Kaiju | *Menzel et al., 2016* | | Metagenomic taxonomy assignment software using unassembled reads |
| Database | Refseq | https://www.ncbi.nlm.nih.gov/refseq/ | RRID:SCR_003496 | Database used with Kaiju for bacterial species assignments of metagenomic reads |
| Software, algorithm | R | *R Core Team, 2019* | RRID:SCR_001905 | Used for statistical analyses |
| Software, algorithm | Matlab-based DLTdv-5 | *Hedrick, 2008* | | Used for video tracking of sourdough height for dough rise profiles |
| Other | Twister PDMS stir bar | Gerstel | | Collection medium for volatile organic compounds in functional assays |
| Other | Lactobacilli MRS agar | Criterion | C5930 | Growth medium for the cultivation of lactic acid bacteria |

*Continued on next page*

*Continued*

| Reagent type (species) or resource | Designation | Source or reference | Identifiers | Additional information |
|---|---|---|---|---|
| Other | CHROMagar Candida | CHROMagar | CA222 | Differential growth medium; creates differential pigmentation and growth phenotypes for distinguishing yeast |
| Strain | *Lactobacillus sanfranciscensis* 17B2 | This paper | MW218985 | |
| Strain | *Lactobacillus brevis* 0092a | This paper | MW218986 | |
| Strain | *Lactobacillus paralimentarius* 0316d | This paper | MW218987 | |
| Strain | *Lactobacillus plantarum* 232 | This paper | MW218988 | |
| Strain | *Saccharomyces cerevisiae* 253 | This paper | MW219042 | |
| Strain | *Wickerhamomyces anomalus* 163 | This paper | MW219039 | |
| Strain | *Kazachstania humilis* 228 | This paper | MW219040 | |
| Strain | *Kazachstania servazzii* 177 | This paper | MW219041 | |

## Sample collection and processing

Sourdough starters were submitted by community scientists participating as part of the Sourdough Project (http://robdunnlab.com/projects/sourdough/). Community scientists were recruited through web sites, social media, and email campaigns worldwide January-March 2017. They were directed to an online Informed Consent form approved by the North Carolina State University's Human Research Committee (IRB Approval Reference #10590). Each participant first answered an extensive online questionnaire consisting of 40 questions related to the source, history, maintenance, and use of their sourdough starter. Upon completion, participants were assigned a unique ID. Participants were instructed to triple-bag ~4 oz of their freshly fed sourdough starter in a new resealable plastic bag, label the bag with their participant ID, and then ship it to Tufts University. In order to preserve participant confidentiality, samples were reassigned a new Sample ID number upon arrival. Each starter sample was subdivided into two subsamples: (1) 1 mL was transferred to a 1.5 mL tube and stored at −80℃ until samples could be processed for DNA sequencing, (2) glycerol stocks (15% glycerol) were made of each sample by combining equal parts sample and 30% glycerol and stored at −80℃ for competition and VOC analyses. In total, we received, processed, and sequenced 560 sourdough starter samples with completed surveys from participants. After quality filtering and rarefying amplicon datasets (see below), 500 were retained.

## Amplicon 16S and ITS rRNA gene and shotgun metagenomic sequencing

To characterize the bacterial and fungal communities of sourdough starters, we followed previously described molecular marker gene sequencing protocols (*Ramirez et al., 2014*; *Oliverio et al., 2017*). In brief, we extracted DNA from 2 mL sub-samples using a Qiagen PowerSoil DNA extraction kit, and then amplified extracted DNA with barcoded primers to enable multiplexed sequencing in duplicate, using the 515 f/806 r for bacteria and ITS1f/ITS2 for fungi. Amplicon concentrations were normalized and sequenced on the Illumina MiSeq platform at the University of Colorado Next Generation Sequencing Facility with 2 × 150 and 2 × 250 bp paired-end chemistry for bacterial and fungal sequencing, respectively. We sequenced multiple DNA extractions and PCR negative controls to check for contamination.

Raw sequences were processed with the DADA2 pipeline (*Callahan et al., 2016*). The DADA2 pipeline detects ASVs as opposed to clustering sequences by percent sequence similarity. Briefly, sequences with N's were removed prior to primer removal with Cutadapt (*Martin, 2011*). Then sequences were quality filtered. For the bacterial sequence data, we used the following parameters: truncLen = 150 for forward reads and 140 for reverse reads, maxEE = 1, and truncQ = 11 and for the fungal data we used the following parameters: minLen = 50, maxEE = 2, truncQ = 2 and maxN = 0. The parameters are different for 16S and ITS due to the variable nature of the ITS region.

After quality filtering, sequence variants were inferred with the DADA2 algorithm, and then we merged paired-end reads. We removed chimeras and taxonomy was determined using the Silva database for bacteria (*Quast et al., 2013*) and the UNITE database (*Nilsson et al., 2019*) for fungi. We then filtered out reads assigned to either chloroplasts or mitochondria from the bacterial taxa table, along with reads that were unassigned at the phylum level. Likewise, we filtered out fungal reads unassigned at the phylum or class levels. We then rarefied the ASV tables to 1260 bacterial reads per sample and 4000 fungal reads per sample and converted to percent relative abundances for downstream analyses. We retained 500 sourdough samples for analyses (e.g. those samples for which we obtained both bacterial and fungal amplicon data).

To obtain high-quality taxonomic species assignments for LAB and AAB and to facilitate comparisons to longer 16S rRNA sequences of isolates, we built phylogenetic trees for both bacterial groups (*Figure 2A-Figure 2—figure supplement 1*). We included high-quality (≥1200 bp) isolate sequences for both groups from the ribosomal database project (*Cole et al., 2014*). In order to preserve continuity between our analysis and previously published studies of sourdough ecology, we refer to the *Lactobacillus* species by their traditional names rather than recently proposed taxonomic reassignments (*Zheng et al., 2020*). To obtain assignments, we created alignments with MUSCLE (*Edgar, 2004*) and then built phylogenetic trees with raxml-HPC (*Stamatakis, 2014*) with the GTR-GAMMA model. We used a patristic distance of 0.97 to assign species taxonomic labels (*Pommier et al., 2009*). ASVs that clustered with two closely related reference sequences were assigned both names. ASVs that are clustered with more than two reference sequences were named by their cluster number, except when reference sequences in a cluster are documented to be closely related, as in the case for the *L. plantarum/L. pentosus/L. fabifermentans* spp. group (*Mao et al., 2015*), which is referred to as the *L. plantarum spp.* group or *L. plantarum*; the *L. casei/L. paracasei/ L. zeae/L. rhamnosus spp.* group (*Dobson et al., 2004*) referred to as the *L. casei spp.* group; the *L. crustorum/L. mindensis/L. farcisminis spp.* group (*Scheirlinck et al., 2007a*), referred to as the *L. crustorum spp.* group; the *A. lambici/A. lovaniensis/A. okinawensis/A. syzygii/A. ghanensis/A. fabarum spp.* group (*Iino et al., 2012*; *Spitaels et al., 2014*), which is referred to as the *A. lovaniensis* spp. group; the *A. orientalis/A. farinalis/A. malorum/A. cerevisiae/A. persici/A. cibinongensis spp.* group (*Li et al., 2014*), referred to as the *A. malorum spp.* group; and the *Gluconobacter wanchernii/G. jabonicus/G. thailandicus/G. cerinus/G. nephelii spp.* goup (*Matsutani et al., 2014*), referred to as the *G. frateurii spp.* group.

We also characterized 40 sourdough starters used for functional analyses (dough rise, VOCs, and sensory notes) via shotgun metagenomic sequencing in order to confirm that the experimental communities revived from glycerol stocks were similar to the original samples. We prepared the samples for shotgun metagenomic sequencing as per *Oliverio et al., 2020* and samples were sequenced on the Illumina NextSeq platform with 2 × 150 bp chemistry at the University of Colorado Next Generation Sequencing Facility. We filtered raw reads with Sickle (*Joshi and Fass, 2011*) with the specified parameters -q 50 and -I 20. On average, each sample consisted of ~3.7 million paired-end reads after quality filtering. We classified the taxonomy of metagenomic reads with Kaiju, using the RefSeq database (*Menzel et al., 2016*).

## Culturing LAB and yeasts

Eight individual isolates of abundant yeasts and bacteria from amplicon data (four yeasts and four bacteria) were isolated from glycerol stocks which were plated onto Lactobacilli MRS agar (Criterion) or Yeast Potato Dextrose (YPD). The yeasts *S. cerevisiae*, *W. anomalus*, *K. humilis*, and *K. servazii* as well as the LAB *L. sanfranciscensis*, *L. brevis*, *L. paramilentarius*, and *L. plantarum* were chosen because of their abundance (within the top five bacteria and yeast in amplicon sequencing, respectively; mean relative abundance). The eight isolates were sourced from eight different starters in our collection. Identities were confirmed using Sanger sequencing of the ITS region for yeast using the primer ITS1f (*Gardes and Bruns, 1993*) and ITS4 (*White et al., 1990*) and 16s regions of bacteria using the primers 27f (*Lane, 1991*) and 1492r (*Turner et al., 1999*). Sanger sequences of the four bacterial isolates were included in the tree used for ASV classification and all clustered with their respective presumed identities' reference taxa (*Figure 2A-Figure 2—figure supplement 1*).

## Growth and pairwise competition assays of LAB and yeast isolates

For growth and competition assays, a liquid cereal-based fermentation medium (CBFM) was made to approximate the dough environment similar to a previously described approach (*Charalampopoulos et al., 2002*). To make CBFM, 100% whole wheat and all-purpose flour were combined in equal proportion (1:1 by mass). The flour mixture was suspended in room temperature (24°C) deionized water (1:9 flour:water by mass) in 500 ml plastic conical centrifuge bottles by shaking for 1 min. This mixture was immediately centrifuged (Beckman GS-6 Series) at x3000 rpm for 30 min (24°C) and the pellet was discarded. The CBFM was then filtered to exclude microbial cells through Falcon disposable filter funnels (0.20 µm pore size).

To quantify growth of each yeast and LAB alone, we standardized input inocula to 2,000 CFUs per 10 µL. Inocula were standardized by diluting 15% glycerol stocks (stored at −80°C) that had a known concentration of CFUs with 1X phosphate buffered saline (PBS). We inoculated 10 µL of each species into 190 µL of CBFM in individual wells of a 96-well plate (n = 5). After cultures grew statically for 48 hr, cultures were homogenized and then transferred 10 µL of the culture into 190 µL of fresh CBFM. We repeated these transfers a total of six times. All incubations were kept at 24°C throughout the duration of the experiment. Total abundance of each species was determined using CFU plating described below.

For competition experiments, yeasts and bacteria were inoculated into wells of a 96-well plate in a fully factorial pairwise design from frozen glycerol stocks (glycerol stocks prepared as described for growth assays). For each inoculum, frozen stocks were plated and counted on either MRS or YPD and standardized to 1,000 CFUs per 5 µL. All reciprocal pairs of each standardized inoculum (5 µL of each member of the pair) were added to 190 µL of CBFM, for a total of 200 µL. After 48 hr of growth at room temperature, cultures were homogenized and 10 µL of each culture was transferred to 190 µL of fresh CBFM. All pairs were replicated five times. A few replicates were lost due to unexpected contamination (*Figure 3—figure supplement 1*). For both growth and competition assays, the number of replicates was chosen based on pilot experiments that demonstrated the extent of variability across replicates.

All replicates were plated and relative abundance of the interacting species was determined after transfers one, three and six. Yeast:yeast pairs were plated on Chromagar *Candida* plates (CHROMagar) at a $10^{-4}$ dilution to differentiate species based on colony morphology, with the exception of pairs containing *Wickerhamomyces anomalus*, which were differentiated on YPD. Bacteria:bacteria pairs were plated at $10^{-5}$ dilutions on MRS where species could be differentiated based on colony morphology. Yeast:bacteria pairs were spot-plated (5 µL of each dilution) on selective media at full to $10^{-5}$ dilutions to quantify CFUs. Selective media were YPD plus chloramphenicol (50 mg/L) to select for yeast and MRS plus natamycin (21.6 mg/L) to select for bacteria. Individual isolates for our experiment were determined to have 'persisted' if they were above the detection limit of 1/100th of the total population at the end of the experiment (transfer six). Co-persistence was defined as both isolates being detectable at that threshold.

## Sourdough experiments to measure functional outputs

To test how distinct sourdough community structures impacted functions, we revived frozen glycerol stocks of 40 starters in a standard flour medium (see medium preparation description below) using a common garden approach. These 40 starters spanned the diversity we encountered in sequencing (*Figure 2A*); we limited our functional analysis to 40 starters due to practical constraints in annotating VOC data. Rather than directly using frozen starters which were shipped to us from community scientists, we first grew up samples in a 'pre-inoculum' that was used to inoculate doughs for functional analysis. We did this to ensure that all cultures were at comparable growth stages prior to inoculation. The flour used was the same mixture used for CBFM (100% whole wheat and all-purpose flour combined in equal proportion 1:1 by mass), but it was prepared differently in order to approximate the moisture content of dough. Flour was autoclaved on a gravity cycle for 20 min to reduce microbial load. Glycerol-stocked communities (200 µL) were added to 1.8 mL sterile distilled water and 2 g autoclaved flour in three replicate communities for each starter pre-inoculum (n = 3 captured the variability we encountered in pilot experiments). Pre-inoculum was mixed with flour and sterile water using a sterile wooden dowel until no dry flour was visible. The mixture was briefly centrifuged in a 15 mL culture tube (to remove dough stuck to the side of the tube walls) and then

incubated for 72 hr at room temperature. Inoculum was created by mixing sterile water (4 mL) and pre-inoculum (~4 g) and was vortexed to homogenize. Autoclaved flour (2 g), water (1600 µL), and inoculum (400 µL) were added to 15 mL falcon tubes and mixed using sterile wooden dowels and then briefly centrifuged to remove from the sides of tubes.

To confirm that the microbial community that formed in these experimental sourdoughs (grown from the 40 starters selected for functional analyses) resembled the sourdough starters from which they originated, we compared microbial community composition between shotgun metagenomic data collected from experimental sourdoughs and the corresponding amplicon sequence data from the original sourdoughs. We calculated Bray-Curtis dissimilarities for the 40 samples from: (1) the initial amplicon data and (2) 16S taxonomic annotations from the shotgun metagenomic data from the experimental samples. We then compared the two dissimilarity matrices with a Mantel test (based on Spearman rank correlation and 999 permutations). We also correlated relative abundances of some of the most dominant taxa including *L. sanfranciscensis*, *L. brevis*, *L. plantarum*, *P. damnosus*, and also the total percent of AAB detected in each sample. Initial and experimental communities were similar in terms of their overall bacterial composition (Mantel $\rho = 0.55$, $p \leq 0.001$) and abundances of common taxa (*L. sanfranciscensis* $\rho = 0.64$, $p \leq 0.001$; *L. brevis* $\rho = 0.50$, $p \leq 0.001$; *L. plantarum* $\rho = 0.70$, $p \leq 0.001$; *P. damnosus* $\rho = 0.47$, $\leq 0.001$; and % total AAB $\rho = 0.75$, $p \leq 0.001$).

## Measuring rates of dough rise

Dough rise was recorded with a Canon EOS 70D DSLR camera set to image every two minutes for 36 hr. The camera was centered at the vertical and horizontal midpoint of each dough batch. Time-lapse photos were compiled, cropped, and enhanced for contrast using Adobe AfterEffects. The top of every dough was tracked through each frame in the MatLab-based digitizing program DLTdv5 (*Hedrick, 2008*). The height of each culture tube ($T_h$) was measured in pixels in Adobe Photoshop. Changes in dough height ($\Delta X$) were normalized to account for distance from the camera and differences in starting height using the following formula: $\Delta X = (X - X_\circ)/T_h$. Distances relative to tube length were converted to absolute distance (mm/hr) by multiplying values by the dimensions of culture tubes (103 mm). Before fitting curves, data was truncated to cut instances where dough fell by more than 5 percent of its maximum height. For each replicate, a logistic growth curve was fitted, and growth rate (r) was calculated with the R package Growthcurver (*Sprouffske and Wagner, 2016*), with a goodness of fit cutoff for r values of $p \leq 0.01$.

## VOC collection and sensory panel

Sourdough starter volatiles were collected using headspace sorptive extraction, which is an equilibrium-driven sample enrichment technique employing a polydimethylsiloxane coated magnetic stir bar (commercially known as Twister, Gerstel). Stir bars (10 mm long x 0.5 mm thick) were suspended in the headspace of each sample using a magnet on the outside of the sample tube cap to suspend the stir bar above the sample. Three replicates of each community were sampled for 24 hr (beginning 12 hr after inoculation). After collection, the stir bars were removed and spiked with 1 µL of 10 ppm ethylbenzene-d10, which was used as an internal standard (Restek). Relative peak areas (analyte/internal standard) served to measure relative concentrations of each compound in the sample. Sterile deionized water and autoclaved flour were analyzed to measure chemical interferences from background samples. Compounds in the starters measured at concentrations less than or equal to concentrations in the water/flour samples were eliminated from the data.

Stir bars were thermally desorbed into the GC column in the gas chromatograph/mass spectrometer (GC/MS, Agilent models 7890A/5975C). The instrument was equipped with an automated multi-purpose sampler (Gerstel) that transferred stir bars to the thermal desorber (TDU)/programmable temperature vaporization inlet (CIS). The TDU (Gerstel) provided transfer of the VOCs from the stir bar to the CIS by heating it from 40°C (0.70 min) to 275°C (3 min) at 600 °C/min, using 50 mL/min of helium. After 0.1 min the CIS, operating in solvent vent mode, was heated from −100°C to 275°C (5 min) at 12 °C/s. The GC column (30 m x 250 µm x 0.25 µm HP5-MS, Agilent) was heated from 40°C (1 min) to 280°C at 5 °C/min with 1.2 mL/min of constant helium flow. MS operating conditions were: 40 to 350 *m/z* at 10 scans/s, 70 eV electron impact source energy, with ion source and quadrupole temperatures of 230°C and 150°C, respectively.

The retention index (RI) for each compound in the sample was calculated based on a standard mixture of $C_7$ to $C_{30}$ n-alkanes (Sigma-Aldrich, St. Louis, MO). Approximately 300 reference standards were purchased from Sigma-Aldrich, Fisher Scientific, Alfa Aesar (Ward Hill, MA), TCI (Tokyo, Japan), Acros Organics (Pittsburgh, PA) and MP Biomedicals (Santa Ana, CA) to confirm compound identity.

Detailed descriptions of data analysis procedures have been previously described (*Kfoury et al., 2018*; *Robbat et al., 2017*). Briefly, Ion Analytics (Gerstel) data analysis software was used to analyze the data. Peak identification was based on the match between sample and reference compound RI and mass spectrum (positive identification) and between sample and compounds found in NIST05/17, Adams Essential Oil Library, and the literature (tentative identification). In instances where no identification was possible, a numerical identifier was assigned such that the data can be used in other metabolomic studies and to capture knowledge of elution and mass spectral profiles should these compounds become important enough to warrant independent collection and analysis. Compound identity is assigned as follows. First, peak scans were required to be constant for five or more consecutive scans ($\leq$20% difference). Second, the level of scan-to-scan variance (SSV) had to be $\leq$5. The SSV represents the relative error of each scan compared to all other mass spectrum scans in the peak. The smaller the difference, i.e., the closer the SSV is to zero, the better the MS agreement. Third, the Q-value was set to $\geq$93. The Q-value is an integer between 1 and 100; it measures the total ratio deviation of the absolute value calculated by dividing the difference between the expected and observed ion ratios by the expected ion ratio, then multiplied by 100 for each ion across the peak. The closer the value is to 100, the higher the certainty of accuracy between sample and reference compound spectrum (positive) or sample and database/library spectrum (tentative identification). Finally, the Q-ratio represents the ratio of the molecular ion intensity to confirmatory ion intensities across the peak; it also must be $\leq$20%. When all of the individual criteria are met, they form a single criterion of acceptance and the software assigns a compound name or numerical identifier.

There were two sets of sensory descriptors that were used in our study: those that were provided by individuals at the time of sample collection (*Figure 1*; *Figure 2—source data 4*), and those that a trained sensory panel used during the functional analysis of a subset of sourdough starters (*Figure 4—source data 2*). We used a simplified system with the larger set of study participants. Moreover, our trained sensory panel was able to distinguish additional sensory notes that we did not anticipate when we sent out the survey to participants. For the initial survey, the aroma characteristics were guided by a list of aroma characteristics commonly used when discussing aroma notes in maltose-based fermented products with consumers. The list was reduced and condensed for survey purposes. For example, 'medicinal' was used to combine the descriptors 'phenolic antiseptic' and 'solvent nail polish,' and the term 'rose' was translated to the more general 'floral.' The descriptor 'musty' was additionally included to account for the 'farmyard' aroma and the potential contribution of VOCs from filamentous fungal growth.

For our detailed functional analysis of the subset of 40 starter samples, sourdough starter aromas were assessed after 36 hr of incubation by a trained and certified descriptive sensory analysis panel from the Tufts University Sensory and Science Center. Samples were coded, randomized, and served blind to the panel one at a time. The panel used modified flavor profile analysis to measure the intensity and characteristics of the aroma in each sample (*Hootman and Keane, 1992*), resulting in primary (dominant) and secondary notes for each sample (*Figure 4—source data 2*).

## Statistical analyses

All statistical analyses were performed in the R environment ('*R Core Team, 2019*," 2012). To assess and visualize overall similarity in sample composition (*Figure 2A*), we hierarchically clustered samples based on pairwise distances in Bray-Curtis dissimilarities (method = ward.d2), as implemented in hclust. Bacteria including LAB and AAB were weighted equally to fungi (yeast only). To test whether geographic location explained any of the variation in observed microbial composition, we ran Mantel tests (method = Spearman with 999 permutations) comparing bacterial and fungal community dissimilarities (Bray-Curtis) to geographic distances (calculated with the Haversine distance from the R package geosphere). We evaluated this relationship for the whole dataset (n = 500) and using continental US samples only (n = 424). We also compared overall fungal to bacterial dissimilarities with Mantel tests for the global and US-only datasets (*Figure 2—figure supplement 4*).

To assess whether abiotic factors including process parameters (data collected from participants in initial survey) and climatic variables (obtained from latitude/longitude of sample submission) were correlated with any of the observed patterns in microbial composition, we performed PERMANOVA tests on bacterial and fungal dissimilarities, including 33 potential predictors (*Figure 2D–E*; *Figure 2—source data 4*): grain base inputs, other process attributes (starter origin, starter input, storage location, number of feeds per month, frequency opened per week, container material, container lid); water source; sensory notes described by participant; pet presence; and location-based climatic variables obtained from latitude/longitude reported (mean annual temperature, maximum temperature, minimum temperature, precipitation, and net primary productivity). We tested pairwise correlations between all continuous predictor variables. Absolute Pearson correlation values were <0.6, except for the following pairs: maximum temperature and temperature (0.8), maximum temperature and seasonality (−0.9), filtered water and tap water (−0.7). Final PERMANOVA models only included significant predictors of community composition (*Figure 2D–E*; *Figure 2—source data 5*). Some survey responses were not completed by all participants and we excluded those missing predictors from our models (thus, n = 426 samples for all models).

For those variables found to explain a significant portion of the variation in overall microbial community composition (for either fungal or bacterial communities), we tested whether particular taxa were enriched under specific conditions (e.g. 'indicator taxa'; *Figure 2D–E*). To test this with continuous variables (for example, starter age or mean annual temperature of starter location) we used Spearman rank correlations to compare to the relative abundances of particular taxa (ASVs). For categorical variables (for example, whether the starter grain base included rye flour 'grain base rye' or the starter storage location) we used indicator correlation indices (r) values as implemented in the indicspecies package (*De Caceres et al., 2016*). For all comparisons, we only included taxa that were detected in ≥10% of samples, and we considered taxa to be indicators if the false-discovery rate (fdr) corrected p-value was ≤0.05. To determine if there are sourdough microbial species that might be restricted to particular regions of the U.S. (*Figure 2B–C*), we used k-means clustering to group samples at two scales: k = 4 (larger regions) and k = 15 (smaller geographic regions) and identified indicator taxa that were significantly (<0.05) enriched in these regions (*Gebert et al., 2018*).

Positive and negative co-occurrence interactions (*Figure 3F*; *Figure 2A*-*Figure 2—figure supplement 3*) were detected using the R package Cooccur (*Griffith et al., 2016*), which uses a probabilistic approach to determine whether, given their abundance in data, species occur more or less often than is expected by chance. Data were transformed to presence-absence of assigned species taxonomy of LAB, AAB, and yeast above a one percent within-sample threshold. Additionally, only species interactions that were predicted to co-occur more than once were considered. To control for multiple comparisons and minimize false positives, Bonferroni-corrected p values are reported. A Monte Carlo simulation was used to determine the likelihood that seven out of eight synthetic co-persistence tests would match our co-occurrence data. We constructed a matrix with the same distribution as that of our significant (p≤0.05) positive and negative co-occurrence interactions; of 16 significant interactions, eight were positive. We randomly drew from that matrix (n = 10,000) to determine the likelihood that the prediction from our synthetic system- that is, that at least seven of a series of eight interactions, would be correct (four negative and four positive interactions, representing the eight interactions observed in our experimental system).

For two functional outputs, dough rise rate and VOC profiles (*Figure 4—figure supplements 1–2*), we first assessed how similar starters were from the same initial inoculum. For VOC profiles, we ran a PERMANOVA to assess how much variation in VOC profiles (represented as Bray-Curtis dissimilarities) was explained by initial starter inoculum. For dough rise, we ran an ANOVA to assess how differences in observed dough rise rates were predicted again, by initial starter inoculum. We then assessed whether overall community composition (of yeast, LAB, or AAB) predicted the composition of VOCs, using mean values within replicates from the same initial inoculum (thus, n = 40) for both VOC profiles and dough rise rates. We used Mantel tests to compare community dissimilarities in VOCs to LAB, AAB and yeast dissimilarities, and also to the Euclidean distances in total % AAB across samples. We also assessed whether any particular taxa were driving differences in the overall VOC composition with Spearman's correlations between taxa relative abundances and the two non-metric multidimensional scaling (NMDS) axes representing dissimilarities in VOC compositions. Likewise, for dough rise we assessed whether any particular taxa were driving differences in the observed rates via Spearman's correlations. For both, taxa were considered significantly correlated if

the p-value$\leq$0.05 (fdr-corrected). Sensory analysis yielded 14 dominant notes across the 40 starters, including yeasty (n = 8), vinegar/acetic acid/acetic sour (n = 9), green apple (n = 6), fermented sour (n = 5), ethyl acetate/solvency (n = 3), and n = 1 sample for alcoholic sour, bready, cheesy, fermented apple, fruity sour, sweaty, toasted corn chips, veggie sulfide, and winey sour. To test whether the dominant sensory note observed from the Tufts Expert Sensory Panel was predicted by % total AAB, we ran a Kruskal-Wallis test with the dominant note as the predictor and used a Dunn test to assess significance of pairwise comparisons. For the sensory analyses (*Figure 4—figure supplement 3*), we only tested notes that were dominant in $\geq$ . 5 samples, thus n = 26 samples rather than 40. We included only the first note reported by the expert sensory panel for our analyses, as this represented the most dominant note.

## Acknowledgements

We are incredibly grateful to the hundreds of community scientists that contributed samples for this project. Esther Miller, Casey Cosetta, Collin Edwards, Ruby Ye, and Emily Putnam provided feedback on earlier versions of this manuscript.

## Additional information

### Funding

| Funder | Grant reference number | Author |
|---|---|---|
| National Science Foundation | MCB 1715553 | Elizabeth A Landis<br>Nicole Kfoury<br>Megan Biango-Daniels<br>Shravya Sakunala<br>Kinsey Drake<br>Albert Robbat<br>Benjamin E Wolfe |
| National Science Foundation | 1319293 | Erin A McKenney<br>Lauren M Nichols<br>Anne A Madden<br>Robert R Dunn |
| National Science Foundation | GRFP | Elizabeth A Landis<br>Angela M Oliverio |
| Cooperative Institute for Research in Environmental Sciences | | Angela M Oliverio |

The funders had no role in study design, data collection and interpretation, or the decision to submit the work for publication.

### Author contributions

Elizabeth A Landis, Angela M Oliverio, Conceptualization, Data curation, Formal analysis, Investigation, Visualization, Methodology, Writing - original draft, Writing - review and editing; Erin A McKenney, Lauren M Nichols, Conceptualization, Data curation, Investigation, Visualization, Methodology, Writing - review and editing; Nicole Kfoury, Megan Biango-Daniels, Data curation, Investigation, Methodology, Writing - review and editing; Leonora K Shell, Anne A Madden, Lori Shapiro, Conceptualization, Data curation, Investigation, Methodology, Writing - review and editing; Shravya Sakunala, Kinsey Drake, Data curation, Investigation, Methodology; Albert Robbat, Resources, Data curation, Supervision, Investigation, Methodology, Writing - review and editing; Matthew Booker, Conceptualization, Writing - review and editing; Robert R Dunn, Conceptualization, Resources, Data curation, Formal analysis, Supervision, Funding acquisition, Investigation, Visualization, Methodology, Project administration, Writing - review and editing; Noah Fierer, Conceptualization, Resources, Formal analysis, Supervision, Funding acquisition, Investigation, Visualization, Methodology, Project administration, Writing - review and editing; Benjamin E Wolfe, Conceptualization, Resources, Supervision, Funding acquisition, Investigation, Visualization, Methodology, Writing - original draft, Project administration, Writing - review and editing

## Author ORCIDs
Elizabeth A Landis ◉ https://orcid.org/0000-0001-9322-3374
Erin A McKenney ◉ https://orcid.org/0000-0001-9874-1146
Anne A Madden ◉ https://orcid.org/0000-0002-7263-5713
Noah Fierer ◉ http://orcid.org/0000-0002-6432-4261
Benjamin E Wolfe ◉ https://orcid.org/0000-0002-0194-9336

## Ethics
Human subjects: Involvement of citizen scientists in this research was approved by the North Carolina State University's Human Research Committee (IRB Approval Reference #10590).

## Decision letter and Author response
Decision letter https://doi.org/10.7554/eLife.61644.sa1
Author response https://doi.org/10.7554/eLife.61644.sa2

# Additional files

## Supplementary files
• Transparent reporting form

## Data availability
All sequence data (both amplicon sequence data of 16S and ITS as well as shotgun metagenomic data) have been deposited in NCBI as BioProject PRJNA589612. All source data for growth curves, interaction assays, VOC profiles, and dough rise data have been deposited in Dryad: https://doi.org/10.5061/dryad.0p2ngf1z1. Metadata (with fields stripped to preserve participant privacy) along with sequence data and taxonomy for the 500 samples reported in this study are available on Figshare: https://doi.org/10.6084/m9.figshare.13514452.v1.

The following datasets were generated:

| Author(s) | Year | Dataset title | | Database and Identifier |
|---|---|---|---|---|
| Landis EA, Wolfe BE | 2020 | The diversity and function of sourdough starter microbiomeshttps:/doi.org/10.5061/dryad.0p2ngf1z1 | Dryad Digital Repository, 10.5061/dryad.0p2ngf1z1 | |
| Oliverio AM, Fierer N | 2020 | Sourdough starter samples | https://www.ncbi.nlm.nih.gov/bioproject/PRJNA589612 | NCBI BioProject, PRJNA589612 |
| Landis EA, Oliverio AM, McKenney EA, Nichols LM, Kfoury N, Biango-Daniels M, Shapiro L, Madden AA, Sakunala S, Drake K, Robbat A, Booker M, Dunn RR, Fierer N, Wolfe BE | 2021 | The diversity and function of sourdough starter microbiomes | https://figshare.com/articles/dataset/The_diversity_and_function_of_sourdough_starter_microbiomes/13514452/1 | FigShare, 10.6084/m9.figshare.13514452.v1 |

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
