## [Decision Letter]

**Acceptance summary:**

There are many anecdotal reports that sourdough starters are strongly differentiated between geographic regions (an example being the sourdough claimed by some to only exist in San Francisco because of its foggy microclimate). This work carries out a detailed analysis of 500 sourdough starters, corroborating both previous findings such as the specific co-occurrence of certain microbes, but also repudiating much of the conventional wisdom about geographic variation in sourdough communities and putting forward new hypotheses, such as the important role of acetic acid bacteria. A result of practical value is that a specific group of bacteria is a good predictor of certain aromas. Finally, it is an excellent example of the value of community science.

**Decision letter after peer review:**

Thank you for submitting your article "The diversity and function of sourdough starter microbiomes" for consideration by *eLife*. Your article has been reviewed by two peer reviewers, including Sara Mitri as the Reviewing Editor and Reviewer #1, and the evaluation has been overseen by Detlef Weigel as the Senior Editor. The following individual involved in review of your submission has agreed to reveal their identity: Rachel Adams (Reviewer #3).

The reviewers have discussed the reviews with one another and the Reviewing Editor has drafted this decision to help you prepare a revised submission.

Our expectation is that the authors will possibly carry out the additional experiments and report on how they affect the relevant conclusions either in a preprint on bioRxiv or medRxiv, or if appropriate, as a Research Advance in *eLife*, either of which would be linked to the original paper.

Summary:

This work carries out a detailed analysis of 500 sourdough starters from both professional and home bakers (with a focus on North America but some samples from other continents). The authors examine the factors that influence the yeast and bacterial communities in sourdough starters, then undertake experimental work to test hypotheses generated by co-occurrence patterns. It also explores correlations between microbial taxa, VOC profiles, aroma, and dough rise rates. The conclusions are a nice combination of corroborations of previous work (specific co-occurrence patterns), disputing of conventional knowledge that sourdough communities vary with geographic distance (the San Francisco sourdough), and new hypotheses to explore (important role of acetic acid bacteria). They also find that one group of bacteria is overrepresented compared to earlier studies, and that the presence or absence of this group is a good predictor of certain aromas. Interestingly, they also show that between-species interactions (succession) may be one of the best predictors of community composition.

The huge effort in data collection and analysis is admirable and therefore gives quite an informative picture on sourdough cultures, particularly in the US, where apparently studies have been somewhat lacking. The paper is well-written, the authors use extensive and robust analytical techniques. The paper was also fun to read and will be interesting to a wide audience.

Essential revisions:

One point that was unclear and may merit further experiments is why the authors did not measure the interactions with acetic acid bacteria (AAB) if they were so surprisingly abundant and had an effect on the aroma. We are guessing this is due to a lack of medium to isolate them on, but this was not very clear. If this is the case, please clarify this point in the text. However, if it is possible to isolate and grow AAB species, these experiments might still be worth doing.

In terms of the analysis:

1) It would be interesting to include a supplementary figure on how many strains of yeast versus bacteria a starter has. Does the balance between yeast versus bacteria act as a predictor for other measured factors?

2) A second suggestion is to explore whether there is a pattern whereby negative interactions are more dominant within kingdom.

3) You state that "The related starters were statistically distinct from other communities, but percent variation explained was quite small." An interesting aspect is how related they were to each other. The fourth paragraph of the Discussion section discusses this, but it's not clear why you could not look at that with the data you have here.

4) Could you use your data to look at similarity between commercial bakers versus home bakers to test the hypothesis in the Discussion? If this is not simple, feel free to leave it for future work.

5) Bray-Curtis was developed to study ecological communities, and was used here on chemicals (subsection “Statistical analyses”). Are the conclusions robust as to the distance metric used? Any support for why this metric is valid for VOC data?

---

## [Author Response]

Essential revisions:One point that was unclear and may merit further experiments is why the authors did not measure the interactions with acetic acid bacteria (AAB) if they were so surprisingly abundant and had an effect on the aroma. We are guessing this is due to a lack of medium to isolate them on, but this was not very clear. If this is the case, please clarify this point in the text. However, if it is possible to isolate and grow AAB species, these experiments might still be worth doing.

While acetic acid bacteria (AAB) were detected across samples at a frequency higher than we expected (given lack of AAB discussions in literature), we did not include them in experimental work for several reasons:

1) We chose to investigate interactions between LAB and yeast given that they are core components of sourdough. AAB, while representing a large component of the bacterial diversity present in some sourdoughs, were not present in all starters (they occurred in 147 of 500 starters at >1% abundance). This is in contrast to the yeast and LAB, which were present across all starters.

2) When we conducted a co-occurrence analysis of all fungal and bacterial taxa detected in the amplicon sequencing, AAB were not significantly associated (either positively or negatively) with other taxa. Of the 16 significant co-occurrence patterns we detected among taxa, none involved any AAB taxa. We only used microbial taxa for our interaction experiments that were significantly associated with other microbes and were among the most abundant taxa.

3) We agree that experimental studies of how AAB impact functional properties of sourdough starters are needed. We cannot complete this work right now due to COVID-19 staffing restrictions and changes in staffing in the lab of our chemistry collaborator. Our current work provides a comprehensive observational dataset that demonstrates how the abundance of AAB could impact sourdough starter communities. We look forward to conducting experiments in the near future where AAB are added to experimental sourdough starters to observe impacts on community composition and functional outputs (dough rise, aromas, etc.).

We have added a brief rationale statement for excluding AAB in our interaction experiments in the Results section.

In terms of the analysis:1) It would be interesting to include a supplementary figure on how many strains of yeast versus bacteria a starter has. Does the balance between yeast versus bacteria act as a predictor for other measured factors?

We agree that this analysis is an interesting and important element of our survey. We have now included the median number of yeasts and bacteria per starter within the text: “starters were composed of a median of three LAB/AAB and one yeast”. We included this supplementary figure as Figure 2—figure supplement 2, a panel which displays histograms of the total numbers of yeast, LAB/AAB, and total numbers of yeast plus LAB/AAB. This figure also includes a plot of yeast richness plotted against bacterial (LAB/AAB) richness. We found a non-significant relationship between LAB/AAB richness and yeast richness (Spearman’s rho P > 0.05). We included this in the figure legend.

This insight from reviewers also spurred us to investigate whether there is a correlation between overall richness of yeasts and bacterial ASVs and richness in volatile organic compounds (VOCs). To address this, we tested whether bacterial and yeast species richness correlates with VOC richness. Again, we found a non-significant relationship: (Spearman’s rho P > 0.05). This suggests that a higher diversity of microbial taxa in a sourdough starter does not lead to higher diversity of VOCs. We added this to the Results section.

2) A second suggestion is to explore whether there is a pattern whereby negative interactions are more dominant within kingdom.

This is an interesting point that we overlooked in our original presentation of the data.

– In our co-occurrence analysis of our amplicon sequencing data, we do find that negative interactions are more dominant within-kingdom. Indeed, all significant interactions (both positive and negative) occur more frequently within-kingdom. Of the 16 significant interactions we detected, only 2 are cross-kingdom interactions: a negative pattern of co-occurrence between *Saccharomyces cerevisiae* and *Lactobacillus sanfranciscensis*, and a positive pattern of co-occurrence between *Kazachstania servazzii* and *Pediococcus damnosus*. This information was added to the subsection “Ecological distributions of sourdough microbes are structured by biotic interactions”.

– In our pairwise interaction experiments, yeast:bacteria species pairs (inter-kingdom interactions) co-persisted with each other in half of all pairings (8 of 16). Yeast:yeast or bacteria:bacteria (within-kingdom) species pairs co-persisted in 3 of 12 pairings (Figure 3B, E). This does demonstrate more positive inter-kingdom interactions than within-kingdom interactions, and we have acknowledged this interesting pattern in the aforementioned subsection.

3) You state that "The related starters were statistically distinct from other communities, but percent variation explained was quite small." An interesting aspect is how related they were to each other. The fourth paragraph of the Discussion section discusses this, but it's not clear why you could not look at that with the data you have here.

We apologize that our analysis of de novo versus commercial starters was not clear in our original submission. We have revised that section of the manuscript to substantially improve clarity (subsection “Geography, process parameters, and abiotic factors are poor predictors of sourdough starter microbiome composition”). As we were re-writing this section, we decided that our analysis of a limited subset of starters (19) did not adequately address whether sourdough starters are stable over time. This should be addressed in another study through experimental approaches. We have much more statistical power with our dataset to address the broader question of whether participants that received their starter from a commercial source (73 participants) had specific microbial taxa compared to those that created their starter de novo or obtained their starter from another individual (417 participants). We therefore removed the analysis of 19 starters from one origin and references to stability in the Discussion.

4) Could you use your data to look at similarity between commercial bakers versus home bakers to test the hypothesis in the Discussion? If this is not simple, feel free to leave it for future work.

This is a very interesting idea, but we only sampled from home bread bakers and not commercial bakers (those working at a bakery). There was some inconsistent use of terms in our first submission (“commercial”, “business”, “individual”) that may have caused confusion. We apologize for that and have made terminology more clear throughout the figures and manuscript.

5) Bray-Curtis was developed to study ecological communities, and was used here on chemicals (subsection “Statistical analyses”). Are the conclusions robust as to the distance metric used? Any support for why this metric is valid for VOC data?

We agree that the Bray-Curtis dissimilarity metric was developed for ecological communities of species, not chemical profiles. But we believe that it is a useful metric for our VOC chemical profile data. One of the main reasons we used the Bray-Curtis metric is that it does not weigh shared zeros in its calculation of similarity, which is important for both types of relative abundance composition data (VOC and relative abundance species composition). The VOC data meets the assumptions for Bray-Curtis, and this metric is commonly for VOC chemical profiles. Below is a sampling of many previously published papers that have used the Bray-Curtis metric for VOC data:

Junker, R.R., 2018. A biosynthetically informed distance measure to compare secondary metabolite profiles. Chemoecology*, 28:* 29-37.

Pascual, J., von Hoermann, C., Rottler‐Hoermann, A.M., Nevo, O., Geppert, A., Sikorski, J., Huber, K.J., Steiger, S., Ayasse, M. and Overmann, J., 2017. Function of bacterial community dynamics in the formation of cadaveric semiochemicals during in situ carcass decomposition. Environmental Microbiology, 19:3310-3322.

von Hoermann, C., Ruther, J. and Ayasse, M., 2016. Volatile organic compounds of decaying piglet cadavers perceived by Nicrophorus vespilloides. Journal of Chemical Ecology, 42: 756-767.

Goelen, T., Sobhy, I.S., Vanderaa, C., Wäckers, F., Rediers, H., Wenseleers, T., Jacquemyn, H. and Lievens, B., 2020. Bacterial phylogeny predicts volatile organic compound composition and olfactory response of an aphid parasitoid. Oikos. In press

Potard, K., Monard, C., Le Garrec, J.L., Caudal, J.P., Le Bris, N. and Binet, F., 2017. Organic amendment practices as possible drivers of biogenic volatile organic compounds emitted by soils in agrosystems. Agriculture, Ecosystems and Environment, 250:25-36.